# Markers of Cellular Proliferation, Apoptosis, Estrogen/Progesterone Receptor Expression and Fibrosis in Selective Progesterone Receptor Modulator (Ulipristal Acetate)-Treated Uterine Fibroids

**DOI:** 10.3390/jcm10040562

**Published:** 2021-02-03

**Authors:** Iwona Szydłowska, Marta Grabowska, Jolanta Nawrocka-Rutkowska, Małgorzata Piasecka, Andrzej Starczewski

**Affiliations:** 1Department of Gynecology, Endocrinology and Gynecological Oncology, Pomeranian Medical University in Szczecin, 71-252 Szczecin, Poland; jolanaw@poczta.onet.pl (J.N.-R.); andrzejstarcz@tlen.pl (A.S.); 2Department of Histology and Developmental Biology, Pomeranian Medical University, Żołnierska 48 Street, 71-210 Szczecin, Poland; martag@pum.edu.pl (M.G.); mpiasecka@ipartner.com.pl (M.P.)

**Keywords:** uterine myoma, ulipristal acetate, proliferation, apoptosis, fibrosis, estrogen receptor, progesterone receptor

## Abstract

There appear to be very few data on the exact mechanisms of a selective progesterone receptor modulator action in myomas. The aim of the study was to assess the effects of ulipristal acetate (UPA) on fibroids, especially on estrogen receptor (ER) and progesterone receptor (PR) immunoexpression, proliferation, apoptosis and tissue fibrosis, and to compare the above parameters in untreated (surgical attention only) and UPA-treated leiomyomas. UPA-treated patients were divided into three groups: (1) good response (≥25% reduction in volume of fibroid), (2) weak response (insignificant volume reduction) and (3) no response to treatment (no decrease or increase in fibroid volume). The study observed a significant decrease in the percentage of collagen volume fraction and ER and PR immunoexpression in the good response group, in the percentage of proliferating cell nuclear antigen (PCNA)- and Ki67-positive cells in the groups with good and weak reactions vs. control group; significantly higher apoptotic index (terminal deoxynucleotidyl transferase-mediated dUTP nick end-labeling (TUNEL)-positive cells) in the good reaction group vs. control group. The results of the study indicate that a good response to UPA, manifested by a volume reduction of myoma, may be associated with a decrease in fibrosis, ER/PR and PCNA and Ki67 immunoexpression and an increase in cell apoptosis within the myoma.

## 1. Introduction

Uterine myomas are benign uterine tumors, frequently developing in patients at reproductive age until menopause. The risk of developing myomas increases with age. It is estimated that fibroids are found in as much as 70% of women above 40 and about 25–30% of them are symptomatic. The most frequent and burdensome symptoms include severe uterine bleeding, pelvic pain, uterine incontinence, constipation and, in patients at reproductive age, infertility or miscarriages [1,2]. In recent years, there have been a number of studies concerning treatment methods of these benign tumors, a vast number indicating uterus-sparing methods. Non-invasive, or at least less invasive, treatment of fibroids is a direct response to growing expectations of the affected people, including improvement of reproductive outcome without radical surgical solutions and without imposing even temporary limits to their life activity.

Uterine growth of myomas is an effect of imbalance between cells’ proliferation and their death. Cellular proliferation itself, however, appears not to be the only contributing factor, as large amount of extracellular matrix (ECM) also plays an important role in the growth of myoma tumors. While non-hormonal factors such as chronic inflammation, repeated damage to uterine muscle and its repair process can initiate formation of myomas, their growth is a result of hormonal stimulation—especially by estrogen and progesterone [3].

Progesterone and its receptors (PR) strictly influence cellular proliferation and, hence, fibroid growth. Estradiol causes an increase in progesterone receptor expression and supports progesterone action in growth of leiomyomas. In comparison with normal myometrium, significantly elevated levels of PR are observed in myomas [4]. The literature demonstrates that in early pregnancy and under hormonal stimulation of estro-progestins hormone replacement therapy (EP-HRT), a rapid growth of myomas occurs.

A selective progesterone receptor modulator (SPRM), ulipristal acetate (UPA), was registered as a drug in preoperative management of symptomatic myomas in order to reduce the size of tumors and eliminate abnormal uterine bleeding associated with fibroids [5,6,7]. An SPRM modulates PR activity as its agonist/antagonist in different tissues. In other words, UPA binds with progesterone receptors in the fibroids and has a PR antagonist action [8]. By changing the receptor signal, SPRMs stimulate apoptosis and inhibit the proliferation of myoma cells by upregulating p21 and p27 proteins, thus prolonging the cell cycle.

Similarly, Ki67 and proliferating cell nuclear antigen (PCNA), being a nuclear molecule in proliferating cells, determine the degree of proliferation. The terminal deoxynucleotidyl transferase-mediated dUTP nick end-labeling (TUNEL) method is applied for detection and quantification of apoptotic cells in myoma tissue.

It has been confirmed that due to the influence of SPRMs, PCNA expression and Ki67-positive cells decrease, a percentage of TUNEL-positive cells increase and anti-apoptotic Bcl-2 protein expression levels reduce [3,9]. Among the proven effects of SPRMs is the reduction in collagen synthesis, resulting in ECM resorption through stimulation of matrix metalloproteinases. These mechanisms lead to myoma shrinkage [9,10,11,12]. Leiomyoma fibrosis, which affects the accumulation of ECM and indicates the inflammatory nature of these tumors, poses interesting research questions, especially when it remains unclear what effect SPRM treatment has on fibrosis process in myomas and whether the accumulation of fibrosis in leiomyoma can affect the response to this treatment. It has been noticed that effects of SPRM action can depend on several factors, as PR expression in tissue and changes in these receptors are conditioned by different ligands [13]. The understanding of estrogen receptor (ER) and PR expression in fibroid tissue post-SPRM hormone treatment remains incomplete.

The aim of the study was to assess the effects of UPA on fibroid tissue, especially on the expression of estrogen and progesterone receptors, proliferative antigens and apoptosis index as well as the amount of tissue fibrosis, and to compare all of the above parameters in untreated, UPA treatment-responsive (significant volume reduction) and little to non-responsive leiomyomas.

## 2. Methodology

### 2.1. Study Groups and Inclusion/Exclusion Criteria

The study was conducted at The Department of Gynecology, Gynecological Endocrinology and Oncology and The Department of Histology and Developmental Biology of The Pomeranian Medical University in Szczecin within a period of four years, between 2015 and 2019.

Samples of leiomyoma tissue were collected from 34 patients at reproductive age, who, in preparation for surgery, received medical treatment with ulipristal acetate (UPA). The UPA-treated group consisted of patients aged 33–55 (mean age 42). Menorrhagia leading to anemia, combined with previously confirmed (by an ultrasound examination) presence of predominantly intramural fibroids constituted a treatment indication for the studied group. All fibroids (type 2; 3; 4; 5; 2–5 according to The International Federation of Gynecology and Obstetrics (FIGO) classification of myomas) were ≤ 10 cm in diameter [14]. Patients with other gynecological pathologies including rapidly growing leiomyomas, abnormal Doppler sonography results, endometrial abnormalities, abnormal PAP smear test results and past or ongoing liver disfunction were excluded from the study.

Patients were treated with 5 mg of ulipristal acetate (Esmya; Gedeon Richter Plc.) administered orally once a day for 3 months. Throughout the duration of therapy, they did not receive any other hormonal treatment. Upon completion of the UPA therapy, a surgery (myomectomy/abdominal supracervical hysterectomy/laparoscopic supracervical hysterectomy) was performed—none of the studied patients presented with cervical pathologies. On the basis of clinical assessment, patients treated with ulipristal acetate were divided into three groups, according to the response to treatment. In patients who responded well (*n* = 20), the volume of myomas decreased significantly, whereas in patients whose response to treatment was weak (*n* = 10), the decrease in fibroid volume was insignificant. In patients who showed no response to treatment (*n* = 4), no decrease or increase in volume of myomas was observed. Reduction in volume of the fibroids by ≥25% was accepted as a criterion for good response to UPA treatment. In the group with good reaction to therapy, a reduction in fibroid volume by 43.2% was observed. The less responsive group (weak reaction) reacted with a decrease in myoma volume by 11.3%.

The control group consisted of twenty myoma specimens obtained from 20 patients at reproductive age (aged 33–51, mean age 41.6) who underwent surgery without prior UPA treatment. Patients in this group were not subjected to hormonal therapy within 6 months prior to surgery. Symptoms such as menorrhagia leading to anemia combined with previously confirmed (by an ultrasound examination) presence of fibroids constituted an indication for surgery (myomectomy/abdominal supracervical hysterectomy/laparoscopic supracervical hysterectomy—none of the studied patients presented with cervical pathologies).

Pelvic ultrasonography was performed using Voluson imaging systems with a 5-MHz probe. Scans were performed by the same ultrasonographer. During transvaginal ultrasound, one to four myomas ranging from 1.5 to 8.5 cm in diameter were found in all patients that qualified for the study and in the control group. Myomas’ volume was measured using an ellipsoid formula (length × width × height × 0.526). In cases with several leiomyomas, the mean volume of two to four myomas was measured. All participating patients underwent transvaginal color Doppler sonography with assessment of fibroid vascularization and velocimetry of the uterine arteries to exclude potential malignant tumors.

Prior to the commencement of the study, the authors received the approval of the Ethics Committee of The Pomeranian Medical University in Szczecin (KB-0012/94/14).

### 2.2. Histological Analysis

In cases presenting with multiple myomas, a biopsy specimen was collected from the largest tumor and then analyzed. Biopsy specimens obtained post-surgery were analyzed for the histological diagnosis of each fibroid. 

Obtained myomas were routinely fixed in 4% buffered paraformaldehyde and were then embedded in paraffin blocks for further analysis. Subsequently, using a microtome, 3-µm thin sections were cut and placed on the polylysine-coated slides.

Sections of the myomas, having been deparaffinized and rehydrated, were stained with standard methods. Hematoxylin and eosin and Mallory’s trichrome staining (in order to assess fibrosis) were performed according to the protocol described in detail by Bancroft and Gamble (2002).

### 2.3. Immunohistochemistry

The laboratory samples were deparaffinized and rehydrated. In order to expose the epitopes, the slides were boiled for 30 min in Target Retrieval Solution Citrate (Dako, Glostrup, Denmark) at pH 6.0 (for PR) and in Target Retrieval Solution (Dako, Glostrup, Denmark) at pH 9.0 (for PCNA, Ki67 and Erα). Next, they were cooled and washed in phosphate-buffered saline (PBS). The endogenous peroxidase was blocked using peroxidase-blocking solution (Dako, Glostrup, Denmark) for 10 min. In order to determine the immunoexpression of the specific proteins, the following antibodies were used: (1) rabbit monoclonal anti-human estrogen receptor α antibody (clone EP1; Dako, Glostrup, Denmark), diluted 1:50; (2) mouse monoclonal anti-human progesterone receptor antibody (clone PgR 636; Dako, Glostrup, Denmark), diluted 1:50; (3) mouse monoclonal anti-human Ki67 antigen antibody (clone MIB-1; Dako, Glostrup, Denmark), diluted 1:100; (4) mouse monoclonal anti-proliferating cell nuclear antigen antibody (clone PC10; Dako, Glostrup, Denmark), diluted 1:100. Antibodies were diluted in Antibody Diluent with Background Reducing Components (Dako, Glostrup, Denmark). The sections were incubated with the primary antibodies in a humid chamber for 30 min. Next, the sections were incubated with a complex containing a secondary antibody conjugated with horseradish peroxidase (Dako, Glostrup, Denmark). Subsequently, diaminobenzidine (Dako, Glostrup, Denmark) was applied. All slides were washed in distilled water, counterstained with Mayer’s hematoxylin (Sigma-Aldrich Co., St. Louis, MO, USA), dehydrated and coverslipped. The slides were examined under a light microscope (Olympus BX 41, Hamburg, Germany). Negative controls for reaction specificity were performed.

### 2.4. TUNEL Assay

Identification of nuclear DNA fragmentation related to apoptosis was performed with the use of the TUNEL assay (terminal deoxynucleotidyl transferase-mediated dUTP nick end-labeling) according to the manufacturer’s protocol (ApopTag^®^ Peroxidase In Situ Apoptosis Detection Kit; Millipore, Billerica, MA, USA). The laboratory samples were deparaffinized, rehydrated and digested with proteinase K (Dako, Glostrup, Denmark). The activity of endogenous peroxidase was blocked with peroxidase-blocking solution (Dako, Glostrup, Denmark) for 10 min. Following this, the slides were incubated with terminal deoxynucleotidyl transferase—TdT (Millipore™, Billerica, MA, USA)—for 60 min in a humid chamber at 37 °C. Subsequently, after washing with PBS, the slides were incubated with the anti-digoxygenin antibody conjugated with peroxidase for 30 min in a humid chamber. To visualize the reaction, diaminobenzidine (Dako, Glostrup, Denmark) was used. Finally, the slides were counterstained with hematoxylin, dehydrated and coverslipped. A negative control was also performed. The sections were examined in a light microscope (Olympus BX 41, Hamburg, Germany).

### 2.5. Quantitative Computer Image Analysis of Mallory’s Trichrome Staining and Immunoexpression of ER, PR, Ki67 and PCNA

With the use of a ScanScope AT2 scanner (Leica Microsystems, Wetzlar, Germany), Mallory’s trichrome-stained and also ER-, PR-, Ki67- and PCNA-immunostained tissue sections were subjected to a scanning procedure at a magnification of 400× (resolution of 0.25 μm/pixel). Next, the obtained digital images of the myomas were analyzed on a computer screen using ImageScope viewer software (Aperio Technologies, Vista, CA, USA).

The quantitative analysis of collagen was performed on slides with Mallory’s trichrome-stained myoma tissue sections, using a positive pixel count algorithm (Aperio Technologies, Vista, CA, USA). Other parameters were set to achieve compliance with the visual assessment of color intensity. The analyzed areas were manually determined. The percentage of collagen positive for Mallory’s trichrome staining was determined in 5 high- power fields for each patient, with an average area of 1.5 mm^2^.

Quantitative analysis of ER, PR, Ki67 and PCNA immunoexpression in the myomas was undertaken; a nuclear v9 algorithm (Aperio Technologies, Inc.) was used. Similarly to the prior analysis, other parameters were set to achieve compliance with the visual evaluation, taking into account the threshold for a positive result—a brown color of the reaction in the cell nucleus. The areas of analysis were also manually determined. Using the algorithm, the percentage of positive nuclei was calculated. The total number of positive nuclei was counted in 5 high-power fields in each patient, with an average area of 1.6 mm^2^.

### 2.6. Statistical Analysis

The results were analyzed using Statistica 13.0 software (StatSoft, Krakow, Poland). Arithmetical means (X), standard deviations (SDs) (X ± SD), medians and range were calculated for each of the parameters. The quantitative values were first analyzed for normality using the Shapiro–Wilk test. The obtained values failed normal distribution assumption; therefore, the non-parametric Kruskal–Wallis test with Dunn’s multiple comparison test for post-hoc analysis and the Mann–Whitney U-test were used respectively to assess the differences between the groups. Differences at *p* < 0.05 were considered to be statistically significant.

## 3. Results

### 3.1. Morphological Studies

In the group with good response to UPA treatment, a statistically significant decrease in fibroid volume was noted (*p* = 0.039)—mean decrease by 43.2%. There were no statistically significant differences in age, body mass index (BMI), gravidity, parity and number of fibroids in groups with good, weak or no response in comparison to the control group (Table 1). Moreover, there were no statistically significant differences in the volume of fibroids before and after treatment in groups with weak or no response (Table 1). Regarding fibroid diameter before and after treatment, there were no statistically significant differences in groups with good, weak or no response (Table 1).

In the control group and the groups treated with UPA, myomas had differentiated histological structures (Figure 1A–D). In these groups, the collagen fractions in myomas were characterized by blue-stained fibers (Figure 1E–H). Statistically significant differences in the percentage of collagen volume fraction in the myomas between UPA-untreated patients and patients with weak or no reaction to UPA were not observed. A significant difference (*p* < 0.05) was only revealed between the control group and the group with good reaction to treatment. There were lower percentages of collagen volume fraction. In myomas with good reaction to UPA, percentages of collagen volume fraction were lower (Table 2).

### 3.2. Quantitative Analysis of ER- and PR-Positive Cells

In the control group and the UPA-treated groups, ER- and PR-positive cells were characterized by brown-stained nuclei (Figure 2). The analysis of the results of ER and PR expression in UPA-treated and non-treated myomas obtained from the relevant groups, established that the percentages of ER- and PR-positive cells were comparable. A significant difference (*p* < 0.001) was revealed between the control group and the group with good reaction to UPA therapy. In the latter, percentages of ER- and PR-positive cells were lower. There was no statistical significance in expression of ER and PR in groups with weak or no reaction to treatment (Table 3).

### 3.3. Quantitative Analysis of PCNA- and Ki67-Positive Cells 

In the control group and the groups treated with UPA, PCNA- and Ki67-positive cells were characterized by brown-stained nuclei (Figure 3). Statistically significant differences in the percentages of PCNA- and Ki67-positive cells in myomas in the group with no reaction to UPA treatment and the untreated group were not found. Significant differences were noted between the control group and the groups with good (*p* < 0.001) and weak (*p* < 0.05) reactions to UPA. There were lower percentages of PCNA- and Ki67-positive cells in myomas with a good or weak reaction to UPA (Table 4).

### 3.4. Quantitative Analysis of TUNEL-Positive Cells 

In the control group and the groups treated with UPA, TUNEL-positive cells (with nuclear DNA fragmentation) were characterized by brown-stained nuclei (Figure 4). Detection of apoptotic cells with the use of the TUNEL assay confirmed that the apoptotic index was significantly higher (*p* < 0.001) in the group with good reaction to UPA treatment than in the control group. The percentages of TUNEL-positive cells were insignificantly higher than in the control group in cases of myomas obtained from the UPA-treated group with weak or no reaction to treatment (Table 5).

## 4. Discussion

There appear to be very few data on the exact mechanisms of action and effects of SPRMs on PR and ER activity in leiomyoma cells. Existing research shows that progesterone increases the production of ECM and induces fibroid growth. ECM activity is conditioned by the presence of PR in uterine fibroids [15]. In animal models, PR expression levels in leiomyoma were described as significantly higher than the expression of ER in tumor tissue [16]. Some studies demonstrate increased numbers of ERs in leiomyomas [17,18], while other argue against it [19,20,21]. In their most recent study, Khan et al. observed that in the tissue of untreated uterine myomas, PR content was significantly higher than that of ER and that gonadotropin-releasing hormone agonist (GnRHa) treatment can significantly decrease both ER and PR expression in fibroids [22]. To date, data on the effects of SPRM on the expression of estrogen and progesterone receptors in myomas remain scanty and somewhat ambiguous. In 2017, Demura et al.—in addition to increased apoptosis and reduced leiomyoma proliferation and angiogenesis induced by UPA—noted no effect on the expression of PRs and ERs in myoma tissue [23]. Tinelli et al. [24], in their study on the effects of UPA treatment on myomas ex vivo, noticed significant differences in the expression patterns of proteins related to regulation of the cell cycle, remodeling of the cell cytoskeleton and also drug resistance. It was revealed that UPA caused a reduction in cofilin—an essential actin regulatory protein—extracellular signal-regulated kinase (ERK) and proto-oncogene tyrosine-protein kinase (Src) phosphorylation and p27 and ezrin protein levels. Additionally, no reductions in protein kinase B (PKB/Akt) phosphorylation and cyclin D1 and β-catenin levels were observed. In vivo and ex vivo findings alike contribute to the general knowledge of biological effects of UPA in treatment of leiomyomas and are essential to the overall understanding of the issue.

Unlike the recent publication by Khan et al., our study demonstrates similar amounts of ERs and PRs in myomas within the control group. Contrary to Demura et al., our research shows a reduction in the amounts of ER and PR in groups subjected to UPA treatment with good response to the therapy compared with myomas without preoperative UPA preparation. The decrease in the number of ER/PR receptors was significantly lower in myomas that were less reactive to UPA in terms of volume reduction. In case of tumors that were non-responsive to UPA, a reduction in ER amount was observed without a noticeable decrease in the amount of PRs. It has to be noted, however, that the number of patients with myomas that were completely non-responsive to UPA treatment was low—hence, conclusions need to be confirmed in further studies. The evaluation of ER/PR levels in response to UPA treatment according to the quality of treatment response itself may possibly be the first study to date assessing data against such criteria.

The existing literature demonstrates that estradiol and progesterone are upregulated, determined by PCNA expression and Ki67 index and cell proliferative activity in fibroids [3,11,25]. It can thus be assumed that PR antagonist effects of UPA cause a reverse reaction to progesterone in treated myomas. There are, however, only a number of studies that attempt to explain the mechanism of UPA action in volume reduction of fibroids. Courtoy GE et al. described that UPA could decrease proliferation, as measured by Ki67-positive cells, stimulate apoptosis and decrease ECM volume in myomas [9]; other studies offered an explanation of mechanisms for the dissolution of ECM [12,26]. The effects of mifepristone, as a progesterone receptor modulator with anti-progesterone activity, were investigated by Chung et al., who established that PCNA expression was significantly reduced in a mifepristone-treated group compared with those in a control group [27]. Similar effects on cellular proliferation and apoptosis were described in uterine myomas after GnRHa treatment [28,29]. In contrast, Yun et al. observed not only elevation in apoptosis but also an increase in proliferation in myomas treated with SPRMs [30].

Our study, through the assessment of two independent parameters, namely PCNA and Ki67 index, confirms reduction in proliferation in leiomyomas after UPA treatment. It demonstrates that higher amounts of these antigens can be found in the fibroid tissue of leiomyomas that weakly respond to UPA treatment in terms of volume reduction. The amount of antigens found is similar to that present in the control group.

We confirm, through the application of TUNEL method, as did Courtoy GE et al. and Yun et al. [9,30], the increase in apoptosis in myomas treated with UPA. The increase is greater in myomas responding well to the drug than in non-responsive ones, in which apoptosis levels are, in statistical terms, insignificantly higher than in the control group.

In recent years, considerable attention has been paid to the fibrotic characteristic of leiomyoma tumors. It has been argued that antifibrotic agents could be a therapeutic target, limiting leiomyoma growth and associated clinical symptoms [31,32]. Khan et al. indicated that in case of the implementation of GnRHa therapy, the occurrence of diffuse fibrosis in myomas may impair the effectiveness of this hormonal treatment [22]. Recent studies proved a dose-dependent inhibition of leiomyoma fibrosis treated with ulipristal acetate [26].

Our study demonstrates significantly lower amounts of fibrosis in myomas treated with UPA. It might be of interest that in cases of myomas not responding or weakly responding to treatment, a greater amount of fibrosis was observed—comparable to those in the control group. This would confirm that Khan’s theory of hormone treatment with GnRH agonist can also be applied to SPRM hormone treatment of myomas.

On 13 March 2020, due to cases of severe liver damage associated with the implementation of UPA, the Pharmacovigilance Risk Assessment Committee (PRAC) issued recommendations to temporarily discontinue any usage of this drug in therapy until further announcement from the European Commission [33]. As of 12 November 2020, the Committee for Medicinal Products for Human Use (CHMP) recommended a restricted implementation of UPA-containing medicines, limited only to treatment of fibroids in premenopausal women with contraindications for surgical treatment (including embolization). As such, this constitutes the final and legally binding decision of the European Commission [34]. The results of our study remain clinically applicable but limited to cases specified in the final statement of the European Commission concerning UPA usage.

Meanwhile, new and promising SPRM options in treatment of uterine leiomyoma have emerged. Vilaprisan, for instance, shows effectiveness similar to UPA. Further clinical trials, especially concerning its safety and efficacy, are currently underway and will be essential for determining conclusions on its clinical usage in the future [35,36]. Recent studies on ER/PR-positive breast cancer tumors suggest significant anti-proliferative activity, measured by Ki67 index, of another SPRM—telapristone acetate [37,38]; however, there are no available data on its implementation in uterine leiomyoma treatment.

Given the same mechanism of drug action in the SPRM group, the above-presented study can potentially be applicable to the new drug mechanism of action in treatment of myomas.

The strength of the study lays in a wide panel of tested parameters and their comparison in groups of myomas, according to the response to UPA treatment. Our research compliments recent studies evaluating the mechanism of SPRM action in myoma tissue.

The relatively small amount of assessed myoma tissue, especially in the group of tumors that were non-responsive to UPA (volume shrinkage), may pose limitations to the study. Nonetheless, we decided to create the group due to the observed changes in the immunohistochemical studies. The above conclusions should be approached with caution and further studies on the subject would be advisable.

## 5. Conclusions

The outcome of UPA therapy is multifactorial and depends on SPRM influence on receptors, proliferation, apoptosis and fibrosis in myoma tissue. The results of the study indicate that a good response to UPA, manifested by a volume reduction of myoma, may be associated with a decrease in fibrosis, ER/PR and PCNA and Ki67 immunoexpression as well as an increase in cell apoptosis in uterine myoma.

## Figures and Tables

**Figure 1 jcm-10-00562-f001:**
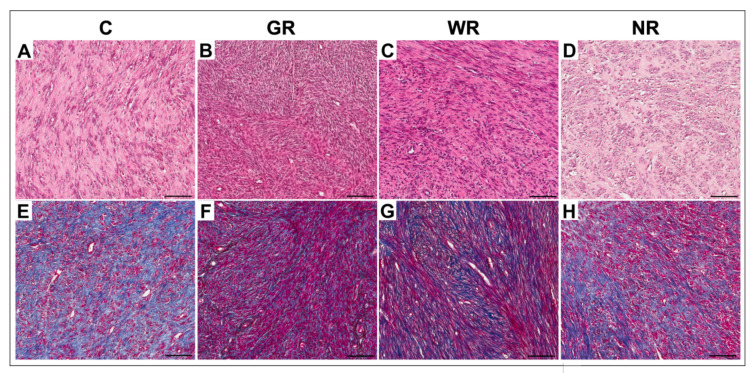
Representative light micrographs of myomas in the control group (**A**,**E**); in the group with good reaction to UPA (ulipristal acetate) treatment (**B**,**F**); in the group with weak reaction to UPA treatment (**C**,**G**) and in the group with no reaction to UPA treatment (**D**,**H**). **A**–**D**: hematoxylin and eosin staining; **E**–**H**: Mallory’s trichrome staining. Scale bar—100 µm. C—control group; GR—good reaction to treatment; WR—weak reaction to treatment; NR—no reaction to treatment.

**Figure 2 jcm-10-00562-f002:**
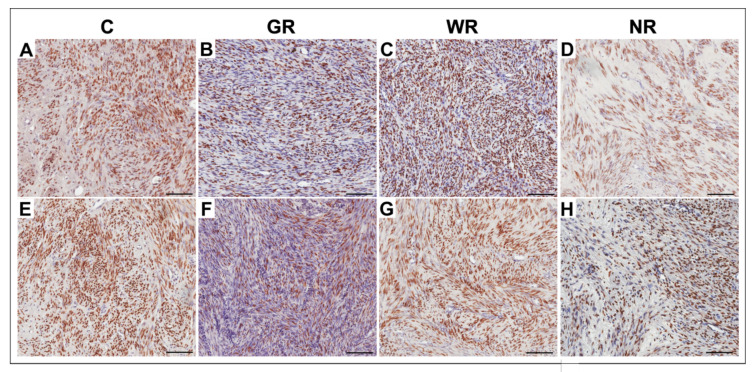
Representative light micrographs of ER and PR immunoexpression in myoma cells (brown-stained nuclei) in the control group (**A**,**E**); in the group with good reaction to UPA (ulipristal acetate) treatment (**B**,**F**); in the group with weak reaction to UPA treatment (**C**,**G**) and in the group with no reaction to UPA treatment (**D**,**H**). **A**–**D**: ER immunoexpression; **E**–**H**: PR immunoexpression. Scale bar—100 µm. C—control group; ER—estrogen receptor; GR—good reaction to treatment; PR—progesterone receptor; WR—weak reaction to treatment; NR—no reaction to treatment.

**Figure 3 jcm-10-00562-f003:**
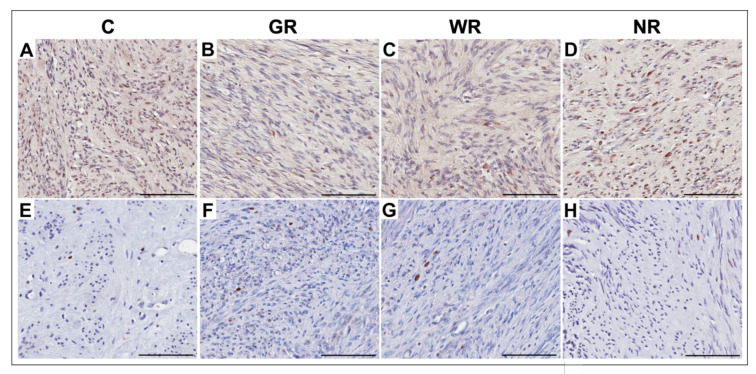
Representative light micrographs of proliferating cell nuclear antigen (PCNA) and Ki67 immunoexpression in myoma cells (brown-stained nuclei) in the control group (**A**,**E**); in the group with good reaction to UPA treatment (**B**,**F**), in the group with weak reaction to UPA (ulipristal acetate) (treatment (**C**,**G**) and in the group with no reaction to UPA treatment (**D**,**H**). **A**–**D**: PCNA immunoexpression; **E**–**H**: Ki67 immunoexpression. Scale bar—100 µm. C—control group; GR—good reaction to treatment; WR—weak reaction to treatment; NR—no reaction to treatment.

**Figure 4 jcm-10-00562-f004:**
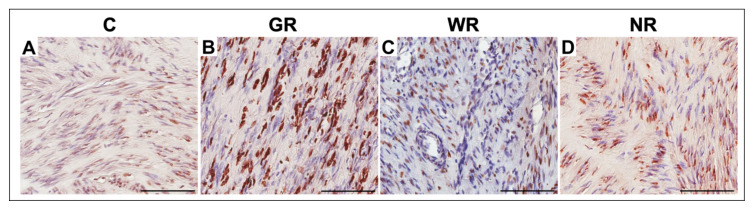
Representative light micrographs of TUNEL-positive myoma cells (brown-stained nuclei) in the control group (**A**), and in the UPA (ulipristal acetate)-treated groups with good (**B**), weak (**C**) and no reaction to treatment (**D**). Scale bar—100 µm. C—control group, GR—good reaction to treatment, WR—weak reaction to treatment, NR—no reaction to treatment.

**Table 1 jcm-10-00562-t001:** Baseline characteristics of the groups.

Group		Control (*n* = 20)	Good Reaction (*n* = 20)	Weak Reaction (*n* = 10)	No Reaction (*n* = 4)
Age	X ± SD	41.6 ± 4.6	42.1 ± 5.3	43.3 ± 5.4	43.0 ± 1.6
	Median (range)	41.5 (33–51)	42.0 (33–55)	46.0 (33–47)	43.0 (41–45)
BMI	X ± SD	24.6 ± 2.3	24.7 ± 2.1	26.0 ± 2.1	26.2 ± 2.1
	Median (range)	24.2 (21.3–30.1)	24.2 (22.2–29.7)	26.1 (22.9–28.7)	25.8 (24.0–29.0)
Gravidity	X ± SD	1.6 ± 0.9	1.4 ± 0.9	1.4 ± 1.0	1.0 ± 0.8
	Median (range)	2.0 (0–3)	1.0 (0–3)	1.5 (0–3)	1.0 (0–2)
Parity	X ± SD	1.2 ± 0.8	1.2 ± 0.7	1.3 ± 0.9	1.0 ± 0.8
	Median (range)	1.0 (0–3)	1.0 (0–2)	1.0 (0–3)	1.0 (0–2)
Number of fibroids	X ± SD	1.5 ± 0.9	1.8 ± 0.8	1.7 ± 0.9	1.5 ± 0.6
	Median (range)	1.0 (1–4)	2.0 (1–4)	1.0 (1–3)	1.5 (1–2)
The volume of fibroids before treatment (cm^3^)	X ± SD	74.0 ± 72.4	81.5 ± 71.7	38.0 ± 61.3	45.9 ± 20.8
	Median (range)	44.9 (5.5–267.8)	49.3 (16.0–297.8)	14.8 (2.9–199.9)	48.5 (18.1–68.4)
The volume of fibroids after treatment (cm^3^)	X ± SD	-	46.3 * ± 41.8	33.7 ± 59.0	67.3 ± 29.4
	Median (range)	-	33.7 (0.0–124.2)	9.1 (1.2–191.8)	74.7 (25.9–94.1)
The diameter of fibroids before treatment (cm)	X ± SD	4.1 ± 1.4	5.1 ± 1.7	4.2 ± 1.8	4.2 ± 1.0
	Median (range)	4.1 (1.7–6.9)	5.2 (1.8–8.2)	3.7 (2.0–7.2)	4.6 (3.1–4.8)
The diameter of fibroids after treatment (cm)	X ± SD	-	4.1 ± 1.8	3.9 ± 2.0	5.0 ± 1.2
	Median (range)	-	4.3 (1.3–6.9)	3.5 (1.4–7.2)	5.4 (3.7–6.0)
Type of surgery		M/ASH/LSH	M/ASH/LSH	M/ASH/LSH	M/LSH

* *p* < 0.05 vs. volume before treatment (Mann–Whitney U test); ASH—abdominal supracervical hysterectomy; BMI—body mass index; LSH—laparoscopic supracervical hysterectomy, M—Myomectomy; X ± SD—arithmetical mean ± standard deviation.

**Table 2 jcm-10-00562-t002:** The percent of collagen volume fraction in fibroids in the control group and in the groups after three months of treatment with ulipristal acetate.

Group	Collagen Volume Fraction (%)	
	Median (Range)	X ± SD
Control (*n* = 20)	43.9 (20.4−78.3)	47.7 ± 16.4
Good reaction (*n* = 20)	32.5 (5.6−77.8)	37.5 * ± 23.6
Weak reaction (*n* = 10)	46.3 (26.1−71.4)	44.4 ± 13.4
No reaction (*n* = 4)	43.1 (22.7−83.3)	46.8 ± 17.7

X ± SD—arithmetical mean ± standard deviation, *—*p* < 0.05 vs. control (Kruskal–Wallis test).

**Table 3 jcm-10-00562-t003:** The percentage of ER-positive and PR-positive cells in fibroids in the control group and in the study groups after three months of treatment with ulipristal acetate.

Group	Estrogen Receptor		Progesterone Receptor	
	Median (Range)	X ± SD	Median (Range)	X ± SD
Control (*n* = 20)	53.2 (33.2−78.8)	54.1 ± 9.6	54.4 (41.3−63.4)	53.8 ± 5.2
Good reaction (*n* = 20)	14.4 (2.5−77.2)	25.4 * ± 20.7	24.9 (2.7−64.1)	28.8 * ± 14.9
Weak reaction (*n* = 10)	47.5 (41.7−63.1)	50.5 ± 7.1	52.5 (40.5−63.9)	52.8 ± 7.5
No reaction (*n* = 4)	35.3 (24.0−81.9)	51.7 ± 25.9	54.3 (33.4−68.2)	53.1 ± 9.2

X ± SD—arithmetical mean ± standard deviation; ER—estrogen receptor; PR—progesterone receptor, *—*p* < 0.001 vs. control (Kruskal–Wallis test).

**Table 4 jcm-10-00562-t004:** The percent of Ki67-positive and PCNA-positive cells in fibroids in the control group and the study groups after three months of treatment with ulipristal acetate.

Group	Ki67		PCNA	
	Median (Range)	X ± SD	Median (Range)	X ± SD
Control (*n* = 20)	2.0 (0.6−3.0)	2.0 ± 0.5	18.0 (10.2−28.7)	18.4 ± 4.4
Good reaction (*n* = 20)	0.4 (0.1−1.9)	0.6 * ± 0.4	10.8 (6.9−17.1)	11.3 * ± 2.6
Weak reaction (*n* = 10)	1.4 (0.6−2.8)	1.5 ** ± 0.5	12.2 (7.0−29.3)	16.2 ** ± 7.0
No reaction (*n* = 4)	1.7 (1.1−2.4)	1.7 ± 0.3	18.0 (13.6−19.6)	17.8 ± 1.9

X ± SD—arithmetical mean ± standard deviation, PCNA—proliferating cell nuclear antigen, * *p* < 0.001 vs. control, ** *p* < 0.05 vs. control (Kruskal–Wallis test).

**Table 5 jcm-10-00562-t005:** The percentage of terminal deoxynucleotidyl transferase-mediated dUTP nick end-labeling (TUNEL)-positive cells in fibroids in the control group and in the study groups after three months of treatment with ulipristal acetate.

Group	Apoptosis	
	Median (Range)	X ± SD
Control (*n* = 20)	12.0 (1.4−32.2)	12.9 ± 7.1
Good reaction (*n* = 20)	25.1 (7.0−40.4)	24.7 * ± 8.5
Weak reaction (*n* = 10)	14.5 (4.9−31.6)	17.5 ± 9.3
No reaction (*n* = 4)	15.0 (3.4−31.3)	16.6 ± 8.8

X ± SD—arithmetical mean ± standard deviation, *—*p* < 0.001 vs. control (Kruskal–Wallis test).

## Data Availability

The data presented in this study are available on the reasonable request from the corresponding author.

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
