# Peer review of "Markers of Cellular Proliferation, Apoptosis, Estrogen/Progesterone Receptor Expression and Fibrosis in Selective Progesterone Receptor Modulator (Ulipristal Acetate)-Treated Uterine Fibroids"

_jcm, 2021, doi:10.3390/jcm10040562_

Round 1

Reviewer 1 Report

The study is very interesting, as a continuation of the previous studies, already published on UPA and fibroids: SzydÅ‚owska I, Marciniak A, Nawrocka-Rutkowska J, RyÅ‚ A, Starczewski A. Predictive Factors of Response to Selective Progesterone Receptor Modulator (Ulipristal Acetate) in the Pharmacological Treatment of Uterine Fibroids. Int J Environ Res Public Health. 2020;17(3):798. Published 2020 Jan 28. doi:10.3390/ijerph17030798; SzydÅ‚owska, Iwona et al. “Predictive Factors of Response to Selective Progesterone Receptor Modulator (Ulipristal Acetate) in the Pharmacological Treatment of Uterine Fibroids.” International journal of environmental research and public health vol. 17,3 798. 28 Jan. 2020, doi:10.3390/ijerph17030798.

The idea of continuing UPA studies is good, but the translation of research into clinical practice is very poor, for the suspension of the use of UPA in the treatment of uterine fibroids due to serious adverse reactions and the decision of Gedeon-Richter to withdraw it from the market.

However, there are some biases in M&M that vitiate study feasibility: 1) The first problem is related to the range of women to be analyzed; if the authors speak of women of reproductive age, the interval chosen between 33 and 55 years (and 33-51 for control group) is not correct. An interval between 30 and 45 years should be inserted, after which the biological and hormonal changes of the patients change significantly. After the patients pass the age of 45 there is a progressive decline in progesterone production; this data negatively affects the homeostasis of a fertile woman.

2) the problem of uterine fibroids mapping in uteri with more fibroids, with poor predictive efficacy if we evaluate the number and diameter in women to be studied and evaluated.

3) measurement of fibroids in women with uteri containing many fibroids is much more accurate with MRI with the same scans, and not with TVUS with manual fibroid detection (operator-dependent and much less standardized).

4) Then there is the histological problem of multiple fibroids, in which to choose, a priori, the largest of the fibroids on which to perform the analysis.

5) in the discussion section, authors affirmed that there are little data on exact mechanisms and effects of UPA on PR and ER activity on leiomyoma cells; but they did not report the last UPA molecular mechanisms on UF samples treated ex vivo with UPA and profiled for drug effects on selected markers. After ex vivo UPA administration, significant changes were observed in the expression levels of proteins related to cell cycle regulation, cytoskeleton remodeling, and drug resistance. The UPA administration reduced cofilin, Erk and Src phosphorylation, p27 and ezrin protein levels, but not Akt phosphorylation and cyclin D1 and β-catenin levels. These findings can explain the biological effects or not of UPA on uterine fibroids (responders/non responders to UPA). Please, add the latest researches.

Other questions: 1) Why were only the intramural symptomatic fibroids chosen and, moreover, without a classification by FIGO of the type of fibroid chosen, and not also the subserosal fibroids which are often equally symptomatic? Why the limit of 10 cm in diameter, given that there are also women candidates for hysterectomy? And why supracervical hysterectomy and not also total hysterectomy?

In the study, to better understand the biological difference between the classes, it would be very useful to divide by age (of the patients treated) all 3 response classes to UPA treatment: good, weak and no reaction.

Reviewer 2 Report

Thank you for your interesting article on cellular markers in ulipristal acetate treated uterine fibroids.

Study group: it is not clear, why one group of patients was treated with esmya whereas the second group was not. What was the criteria for treatment with esmya?

A table with patient characteristics is missing (should include size of myomas)

Discussion: What is the relevance for a clinician especially as ulipristial acetate is not on the market anymore?

Round 2

Reviewer 1 Report

The authors correctly answered the reviewers' questions, inserting adequate and correct comments. Now, after this revision, the manuscript can be accepted in its present form.

Author Response

Reply 1: Thank you very much for your precious comments and suggestions that undoubtedly improved the quality of the submitted manuscript.

Reviewer 2 Report

Thank you for addressing my comments and for making the necessary changes.

The one thing still missing in my opinion is one clear table of patient characteristics.

It is not necessary to add the age in each table. One table in the beginning of the results section, including age, BMI, Gravidity, Parity, Myoma size, type of surgery (hysterectomy, myomectomy, etc..) or any other relevant piece of information you might have regarding patient history, would suffice and would allow the reader to get a proper overview of your study population and to see how the groups compare in regards baseline characteristics.

Author Response

Reply 1: Thank you very much for your precious comments and suggestions that undoubtedly improved the quality of the submitted manuscript. According to your suggestion we have prepared one table including age, BMI, gravidity, parity, myoma size, and type of surgery, to get a proper overview of the study population. In the place of previous tables 1 and 2 we have  inserted new table 1 with all necessary data.